# Healthy Lifestyle Motivators of Willingness to Consume Healthy Food Brands: An Integrative Model

**DOI:** 10.3390/foods14010125

**Published:** 2025-01-04

**Authors:** Elizabeth Emperatriz García-Salirrosas, Manuel Escobar-Farfán, Jorge Alberto Esponda-Perez, Miluska Villar-Guevara, Rafael Fernando Rondon-Eusebio, Ghenkis Ezcurra-Zavaleta, Elena Matilde Urraca-Vergara, Mauricio Guerra-Velásquez

**Affiliations:** 1Grupo de Investigación e Innovación para el Emprendimiento y Sostenibilidad, Universidad Nacional Tecnológica de Lima Sur, Lima 15816, Peru; 2Department of Administration, Faculty of Administration and Economics, University of Santiago of Chile (USACH), Santiago 9170020, Chile; manuel.escobar@usach.cl (M.E.-F.); mauricio.guerra@usach.cl (M.G.-V.); 3Faculty of Nutrition and Food Sciences, Universidad de Ciencias y Artes de Chiapas, Tuxtla Gutiérrez 29000, Mexico; jorge.esponda@unicach.mx; 4Escuela Profesional de Administración, Facultad de Ciencias Empresariales, Universidad Peruana Unión, Juliaca 21100, Peru; miluskavillar@upeu.edu.pe; 5Department of Humanities, Universidad Privada del Norte, Lima 15314, Peru; rafael.rondon@upn.edu.pe; 6Facultad de Ciencias Económicas, Escuela de Administración, Universidad Nacional de Tumbes, Tumbes 24001, Peru; gezcurraz@untumbes.edu.pe; 7Programa de Estudios de Ingeniería Industrial, Facultad de Ingeniería, Universidad Privada Antenor Orrego, La Libertad 13000, Peru; eurracav@upao.edu.pe

**Keywords:** healthy lifestyle motivators, healthy food consumption, self-identity, attitudes, moral norms, consumer behavior

## Abstract

This study evaluated how healthy lifestyle motivators (MHLs) influence the Peruvian market’s willingness to consume healthy food (WCHBF). The main objective was to analyze the relationship of variables, such as attitude (ATT), perceived behavioral control (PBC), self-identity (SI), and moral norms (MN) with the WCHBF. This study adopted a quantitative, non-experimental, and cross-sectional approach, using a self-administered questionnaire for data collection. A total of 585 individuals participated. The participants were consumers of the Unión brand, which specializes in healthy food. A 5-point Likert scale was used to evaluate the key variables. For the analysis, IBM SPSS Statistics 25 programs were used to examine the demographic data and SmartPLS 4.1.0.9 was used to assess the conceptual model using partial least squares (PLS-SEM). The results showed that healthy lifestyle motivators positively influence the willingness to consume healthy food and ATT, PBC, SI, and MN variables. In turn, these variables significantly impact the willingness to purchase healthy food. The findings suggest that marketing campaigns should highlight these motivators to encourage the consumption of healthy food. The implications of this study reinforce the importance of understanding psychological factors in consumer decision-making.

## 1. Introduction

The demand for healthy food consumption increased considerably after COVID-19 [1]. Healthy product brands have stepped up their efforts to offer various healthy eating options, which are becoming more enjoyable and can reach a more significant number of followers [2,3]. These brands offer products enriched with nutrients that have been scientifically proven to have the ability to prevent chronic diseases [4,5,6]. Recent studies have revealed that a healthy lifestyle will positively affect various dimensions of human life [7,8]. Therefore, consuming nutritious food translates into sustainable diets that promote good health, comprehensive well-being, and sustainability [9,10].

Eating healthy brands of food plays a vital role in preventative eating and promoting a healthy lifestyle, as these products typically offer options with good nutritional value, such as low levels of sugar, saturated fat, and sodium, and many important nutrients, such as fiber, vitamins, and minerals [11,12]. A healthy brand also focuses on transparency and reducing additives, which minimizes the risk of chronic diseases, such as obesity, diabetes, and heart disease. Additionally, many health brands focus on sustainable and ethical practices that benefit not only the health of consumers but the environment, which has a positive impact on nutrition and society at large [13,14,15].

Following this trend, consumers have been improving their eating habits, prioritizing health, and even going against their behavioral inclinations [9,16]. Previous studies validate the intense interest in reflecting before a purchase that could involve the health of people and their families [17,18,19]. They are increasingly willing to abandon their bad habits and change them for more sustainable ones. In addition to all this, consumers feel a greater desire to increase their life expectancy and improve their quality of life, which has led to a significant growth in the demand for more functional food that offers more important benefits than conventional ones [15,20].

In Peru, food and health problems are complex and are characterized by the coexistence of malnutrition and obesity. In rural areas, malnutrition continues to experience high rates; while in urban areas, overweight and obesity are suddenly increasing, due to the excessive consumption of processed food and alcohol, as well as a sedentary lifestyle [21]. This has led to an increase in non-communicable diseases, such as type 2 diabetes and high blood pressure. At the public policy level, programs such as warning labels on products containing fat, sugar, and sodium have been implemented, as well as programs to promote physical activity in both public and private institutions and educational centers, such as schools, institutes, and universities, although the challenge remains a permanent change in people’s behavior and purchasing decisions [22].

The background mentioned above reveals a growing interest in continuing to study these topics from the perspective of academics, health professionals, and the business sector. A review of the bibliometric indicators shows that the ten countries that report the most scientific results are the United States, China, the United Kingdom, Australia, Canada, Germany, India, Turkey, Spain, and Italy. These countries have developed their studies in various areas, sectors, and populations, such as medicine, social sciences, psychology, and business. When examining scientific dissemination by country, no study was found in the Peruvian context that guides the topics in question, making this research valuable to fill the knowledge gap and contribute significantly to the academic community and professionals in the sectors involved. Considering that Peru has great cultural, ethnic, and socioeconomic diversity, values, norms, and beliefs vary considerably between urban and rural regions and communities. This could affect how people perceive and react to certain behaviors, especially if this translates into healthy food choices. A study involving these topics would help design more effective interventions to promote healthy and sustainable eating practices in the Peruvian population. In this sense, the main objective was to analyze the relationship of variables, such as attitude (ATT), perceived behavioral control (PBC), self-identity (SI), and moral norms (MN), with the willingness to consume healthy food (WCHBF).

## 2. Literature Review and Hypothesis Development

Health perceptions and beliefs determine dietary decisions [23]. In this context, a healthy and practical lifestyle leads to products that correspond to these values [24], such as food without added sugars or preservatives and from the environment [25]. In addition, when consumers see that a brand is promoting a product that supports their health goals, it generates greater trust [26] and loyalty toward that brand [27,28], which makes it easier to make good decisions and more likely to repeat a choice in favor of healthy alternatives [29]. Schubert [18] conducted a study arguing that positive motivations lead to healthier eating and more proportional results in delicious and healthy food. In that sense, decreasing the motivation to consume healthy food through reappraisal, a sedentary lifestyle, excessive consumption of ultra-processed food, and lack of attention to mental well-being reduces the motivation to consume healthy food. Therefore, the following hypothesis is proposed:

**H1.** 
*Motivators for a healthy lifestyle significantly influence the willingness to consume healthy branded food.*


Motivators for a healthy lifestyle, such as the pursuit of physical well-being, disease prevention and improving quality of life, condition dietary decisions [15,25,29,30]. A study of Dutch, Turkish, and Moroccan residents reveals that personal health was the most common motivator for a healthy lifestyle [25]. When people are motivated to stay healthy, they develop positive attitudes toward products that they believe meet their health goals, with healthy brands being an extension of these principles, offering nutritious and sustainable products [10]. This combination of motivators and brand offerings not only promotes positive attitudes toward healthy eating but encourages purchasing behaviors in line with a lifestyle that prioritizes health [11]. Therefore, a positive attitude toward healthy food brands translates into a filter that guides purchasing decisions based on the congruence between personal values and what the market offers [12]. Some studies [13] also report that perceptual, personal, and emotional motivators are strongly linked to attitude. Attitude is a critical element in the decision-making process, and previous research maintains that it can act as a mediator in the relationships between behavioral intention, thus conditioning its occurrence. Based on this solid base of empirical evidence, the following hypothesis is proposed:

**H2.** 
*Motivators for a healthy lifestyle significantly influence attitudes toward healthy branded food.*


When driven by physical, mental, or emotional well-being goals, people feel more in control and direct their behavior toward healthy habits that could be closely associated with the prevention of various diseases, including chronic ones [31,32]. When a person is motivated to be healthy, they have more control over their behavior because they make wiser decisions, plan better, and develop conscious habits to live longer and better [33]. This control allows you to feel more confident in the face of obstacles (such as lack of time, financial shortcomings, weather conditions, or food temptations), facilitating the establishment and maintenance of good habits, such as eating healthily, exercising regularly, and coping with stress [34,35]. Thus, motivational factors increase the desire to change and reinforce the feeling that they have the tools and the ability to achieve those goals. Studies have maintained that self-discipline and self-control play a predominant role in a healthy lifestyle [36,37,38]. Considering these statements, the following hypothesis is proposed:

**H3.** 
*Motivators for a healthy lifestyle significantly influence perceived behavioral control.*


When people adopt healthy habits, such as regular exercise, adequate rest, and a nutritious diet, these behaviors become essential to their self-concept [39,40]. In that sense, self-identity is created through everyday actions and decisions. If a person is motivated to be healthy, these qualities define who they are [39]. For example, people who prioritize healthy eating and their overall well-being may perceive themselves as disciplined, responsible, and committed to their health, reinforcing their identity around those values [41]. Moreover, the motivational factors of a healthy lifestyle affect not only the inner perception but the way others perceive that person. By having good habits, people can create an image related to self-care, stability, and responsibility, which can affect their social identity. The connection between health goals and self-identity allows people to live a healthy life, efficiently developing their behavior and perceiving themselves as someone who prioritizes and values well-being [40]. In this regard, a recent study suggested that marketing supervisors and brand managers should focus on personal attributes rather than social factors to promote healthy lifestyles and encourage healthy consumption [39]. In this context, the following hypothesis is proposed:

**H4.** 
*Motivators for a healthy lifestyle significantly influence self-identity.*


Motivators for a healthy lifestyle significantly influence moral norms. Previous studies have established that health concerns, disease prevention aspirations, general well-being, environmental awareness, and ethical considerations are potent motivators in selecting healthy food and adopting healthy lifestyles. These motivators not only influence direct consumption decisions but contribute to forming positive attitudes toward a healthier lifestyle [3,42,43,44,45]. Furthermore, Honkanen et al. [46] and Habib et al. [47] demonstrated how motivators for a healthy lifestyle are closely linked to forming and strengthening moral and ethical norms. Their study reveals that consumers motivated by health and sustainability tend to internalize these values as part of their ethical code, influencing their future food choices. Previous research has demonstrated how health motivators significantly contribute to forming positive moral attitudes toward sustainable food choices and dietary patterns. Studies have found that individuals with health-driven solid motivations tend to develop more conscious moral attitudes toward organic food and plant-based diets [48,49]. Similarly, Enriquez and Archila-Godinez [50] found that individuals who exhibit stronger motivation toward a healthy lifestyle tend to develop more rigorous ethical norms regarding responsible consumption and personal care. This positive impact of health motivators on moral attitudes gives us hope for a healthier future. Therefore, the following hypothesis is proposed:

**H5:** 
*Motivators for a healthy lifestyle significantly influence moral norms.*


The role of consumer attitudes in purchasing decisions is a crucial area of study in the academic literature. Various researchers have found evidence supporting this connection across different food market contexts [51,52,53]. In functional food, Küster-Boluda and Vidal-Capilla [3] demonstrated that positive consumer perceptions toward these products translate into a higher likelihood of consumption. This finding highlights the crucial role of consumer attitudes in influencing purchasing decisions. Similarly, in the organic product sector, Khan et al. [54] observed that a positive attitude generates purchase intentions and can lead to the actual acquisition of the product. This reinforces that consumer attitudes are crucial for purchasing decisions [39,55].

The research of Roseman et al. [56] emphasizes the relevance of understanding the relationship between consumer attitudes toward food and purchase intentions. In this context, fostering a positive perception among consumers is fundamental, as it can significantly impact their food purchasing and consumption patterns [39]. Other studies, such as those conducted by Canova et al. [57] and Lim and Goh [58], have corroborated these findings, establishing a consistent correlation between consumer attitude and the intention to consume health-oriented products across various cultural and geographical contexts. Based on this solid foundation of empirical evidence, the following hypothesis is proposed:

**H6.** 
*Attitude significantly influences the willingness to consume food from healthy brands.*


The theory of planned behavior introduces the concept of perceived behavioral control, crucial in the consumers’ predisposition toward purchasing healthy food products [23,53]. Previous studies have demonstrated a greater tendency to consume food from healthy brands when individuals exercise control over their consumption habits [59,60,61,62,63]. Research has revealed a notable relationship between the intention to purchase organic food and the perception of mastery over one’s actions. This finding underscores the relevance of self-confidence in making conscious decisions about nutrition [53,61]. Similarly, Aliaga-Ortega et al. [62] found that consumers are more likely to purchase their preferred food when they perceive more significant control over their behavior. However, this perception of control may fluctuate depending on circumstances, which could be related to the degree of autonomy consumers experience when selecting and acquiring their food [64]. Taking these precedents into consideration, the following hypothesis is proposed:

**H7.** 
*Perceived behavioral control significantly influences the willingness to consume food from healthy brands.*


Recent studies have shown that a strong self-identity as a health-conscious consumer is linked to the intention to buy healthy food and follow through with those purchases. These findings remain consistent even when other relevant psychological and social factors are considered as a theory of planned behavior [1,65]. McCarthy et al. [66] highlighted the crucial role of identity in forming and maintaining healthy eating behaviors, arguing that the internalization of an identity associated with healthy eating acts as a self-regulatory mechanism that guides food choices. Similarly, Qasim et al. [67] demonstrated that environmental self-identity, closely related to healthy consumer identity, is a crucial mediator between consumer values and purchase intentions for organic and healthy food. This mediating role directly affects purchase intentions and amplifies the effect of other consumption values on eating behavior [68,69]. Reinforcing these findings, Khare and Pande [70] revealed that a green self-identity significantly influences trust toward organic food retailers and, by extension, a willingness to consume products from brands perceived as healthy. In this context, self-identity is an internal motivating factor influencing consumer preferences and choices [39]. Therefore, the following hypothesis is proposed:

**H8.** 
*Self-identity significantly influences the willingness to consume food from healthy brands.*


Moral norms significantly influence the willingness to consume food from healthy brands, as these norms, rooted in individual ethical values and principles, operate beyond external social pressures [71,72,73]. Canova et al. [57] and Issock et al. [74] examined the role of moral norms in sustainable purchase intentions, specifically in the context of fair-trade products. Their findings demonstrate that moral norms play a crucial role in shaping purchase intentions for ethically produced products, which can be extended to food from healthy brands often perceived as more ethical and sustainable. In the specific realm of organic food, Saleki et al. [75] investigated the factors driving Malaysian consumers’ purchase intentions. Their study revealed that moral norms significantly predict the intention to purchase organic foods, reinforcing that ethical and moral considerations play a fundamental role in healthy food consumption decisions. For their part, Carfora et al. [76] integrated the theory of planned behavior, value-belief-norm theory, and the concept of trust to explain consumers’ intention to purchase natural food. Their results underscore the importance of moral norms in shaping purchase intentions for natural food, which are often associated with healthy brands. Lastly, Yazdanpanah and Forouzani [77] applied the theory of planned behavior to predict Iranian students’ intention to purchase organic food. Their study demonstrated that moral norms are a significant factor in predicting these intentions, suggesting that ethical and moral considerations transcend cultural boundaries in the context of healthy eating.

**H9.** 
*Moral norms significantly influence the willingness to consume food from healthy brands.*


Figure 1 shows a graphic representation of the theoretical model according to the hypotheses raised.

## 3. Materials and Methods

### 3.1. Context

The present study follows a quantitative approach, focusing on measuring and analyzing the influence of healthy lifestyle motivators (MHLs) on several intermediate variables: attitude (ATT), perceived behavioral control (PBC), self-identity (SI), and moral norms (MN). It evaluates how these variables affect the individuals’ willingness to consume healthy branded food (WCHBF). This approach allows us to understand the underlying dynamics between motivators toward healthy lifestyle habits and the individuals’ consumption decisions.

A non-experimental cross-sectional design was used, which implies that the data were collected at a single point in time without direct intervention or manipulation of the variables. The model proposed in this research is based on structural equation analysis (SEM), explicitly using the partial least squares technique (PLS-SEM). This methodology is ideal when seeking to maximize the explained variance [78,79] and when complex models with multiple relationships between latent constructs are analyzed [80].

For data collection, 620 surveys were applied using the Google Forms platform. Only those responses in which participants gave their informed consent were considered valid. Of the surveys distributed, 585 were fully completed and used for analysis. The sampling method was non-probabilistic by convenience, targeting young university students, a population relevant to this study due to their active interest in healthy food consumption. The sample size adequacy was determined based on the guidelines provided by Hair [81], which outlines the minimum sample sizes required for different levels of path coefficients and statistical power. For a minimum path coefficient (pmin) ranging from 0.11 to 0.2, a significance level of 5%, and a statistical power of 80%, the minimum required sample size is 155. With 585 respondents, this study’s sample size far exceeds this minimum requirement, ensuring robust statistical power and reliability in the results.

According to Table 1, the sociodemographic results mostly comprised women (64.96%) and young people (91.97%). Most are single (92.99%) and are studying careers related to Health Sciences (69.57%). As for religion, 81.20% identify themselves as Adventist. In terms of academics, 89.23% are at the undergraduate level.

### 3.2. Measures

This study employed a comprehensive measurement instrument consisting of 24 items across six dimensions (Appendix A). Healthy Lifestyle motivators (HLB) were assessed using Downes’ seven-item scale [15]. Attitude (ATT), perceived behavioral control (PBC), self-identity (SI), and moral norms (MN) were measured using Yazdanpanah and Forouzani’s ten-item questionnaire [77], with items distributed as follows: three for ATT, three for PBC, two for SI, and two for MN. Finally, a willingness to consume healthy food (WCHBF) was evaluated using seven items adapted from Küster-Boluda and Vidal-Capilla’s questionnaire [3]. All items are evaluated using a 5-point Likert scale, where “1” means “Totally disagree” and “5” means “Totally agree”. The digital questionnaire was divided into two sections. The first section presented the items already mentioned, and the second section consisted of questions related to sociodemographic data, such as age, sex, and marital status.

### 3.3. Statistical Analysis

For the present study, a statistical analysis was carried out in two complementary phases, descriptive and structural, based on the method of structural equations using partial least squares (PLS-SEM). In the first phase, IBM SPSS version 26 software was used for descriptive analysis of the study variables. In the second phase, SmartPLS version 4 software was used to perform structural equation analysis (PLS-SEM). The analysis included evaluating both the measurement model and the structural model.

The measurement model was evaluated to ensure the reliability and validity of the latent constructs. Composite reliability (CR) was used, requiring a value greater than 0.70 to ensure the internal consistency of the items. In addition, Cronbach’s alpha, with a threshold of 0.70, was used to indicate high reliability in the scales (Vinzi et al., 2010) [82]. For convergent validity, average variance extracted (AVE) was used, accepting values equal to or greater than 0.50, which indicates that, on average, the items explain more than half of the construct’s variance [78]. Discriminant validity was assessed using the Fornell–Larcker criterion, ensuring that the square root of the AVE of each construct was more significant than the correlations between that construct and the others, confirming the distinction between the constructs [83,84].

The structural model was evaluated to analyze the path coefficients, coefficient of determination (R^2^), and statistical significance of the path coefficients. The R^2^ metric, which indicates the proportion of variance explained in the endogenous constructs, serves as a key measure of the explanatory power of the model. In social science research, R^2^ values are usually interpreted as substantial, moderate, or weak when they reach thresholds of 0.75, 0.50, and 0.25, respectively. This interpretation is in keeping with the nature of this field, where numerous external factors influence human behavior and attitudes, so it is common for models to yield lower R^2^ values than those observed in the physical sciences or engineering disciplines. Significance is confirmed with *p* values < 0.05 [85].

## 4. Results

Table 2 shows the results of the reliability and validity analysis of the measurement model, evaluating the factor loadings, Cronbach’s alpha, composite reliability (rho_a), and average variance extracted (AVE) for each variable. Regarding the factor loadings, most exceed the threshold of 0.70, indicating good compensation between the items and their respective constructs. The values of Cronbach’s alpha and composite reliability (rho_a) are higher than 0.70 for all variables, demonstrating high internal consistency. Regarding the average variance extracted (AVE), all variables meet the threshold of 0.50, which ensures adequate convergent validity. The results in the table indicate that the model’s constructs meet the criteria of reliability and convergent validity, confirming the robustness of the measurement model.

Table 3 shows that the square root of each construct’s AVE (values on the diagonal) exceeds the correlations with the other constructs, thus fulfilling the Fornell–Larcker criterion. For example, the square root value of the AVE for “Attitude” (ATT) is 0.962, which is greater than all its correlations with the other constructs (MHL, MN, PBC, SI, WCHBF). Similarly, the other constructs also fulfill this condition, which ensures the discriminant validity of all the constructs.

Another method to evaluate discriminant validity in structural equation models is cross-loading analysis. According to Chin [86], to comply with discriminant validity, items must load higher on their associated construct than on any other. In the case of the present study, the highest loadings are on the ATT items (ATT1, ATT2, ATT3) on their associated construct, with values above 0.95. By contrast, the loadings on other constructs are significantly lower (range 0.538 to 0.670). This confirms that ATT has discriminant validity. Similarly, the MHLs, MN, PBC, SI, and WCHBF items have their highest loadings on their respective constructs, meeting the discriminant validity criterion (Table 4).

To complement the discriminant validity analysis, the heterotrait–monotrait ratio (HTMT) analysis was additionally performed. Under this criterion, to meet discriminant validity between constructs, HTMT values should be below 0.85 or 0.90 in some cases [87]. In the present work, the HTMT values between constructs are all below 0.85, except PBC with MN (0.871), but they are still close to the 0.85 threshold, so they are considered acceptable. The other values, such as MHLs with ATT (0.728) and WCHBF with ATT (0.593), essentially meet the criterion, confirming discriminant validity (Table 5).

Table 6 and Figure 2 present the results of hypothesis testing using structural equation modeling (PLS-SEM). Each hypothesis was evaluated based on the original sample mean (O), the sample mean (M), and the associated “*p*-values”, which must have values less than 0.05 to be considered significant (*p* < 0.05). The results show that Healthy Lifestyle Motivators (MHLs) have a positive and significant influence on the willingness to consume WCHBF-branded healthy food (H1) (coefficient 0.135, *p* = 0.004). Although the effect is moderate, it indicates that people with motivators associated with a healthy lifestyle are more predisposed to consume this type of product. Likewise, MHLs have been found to have a strong positive influence on attitude (ATT) toward healthy food (H2) (coefficient 0.692, *p* < 0.001), suggesting that these motivators are determinants in how healthy food are perceived. It has also been found that MHLs also significantly influence perceived behavioral control (PBC) (H3) (coefficient 0.599, *p* < 0.001), highlighting the perception of ability or control when consuming healthy food. Likewise, the positive relationship with self-identity (SI) has been tested (H4) (coefficient 0.601, *p* < 0.001), which shows that healthy lifestyle motivators are related to how people see healthy consumption as part of their identity. Additionally, it has been found that motivators also have a positive effect on moral norms (MN) (H5) (coefficient 0.623, *p* < 0.001), which implies a connection between these motivators and the ethical sense related to healthy consumption.

The results also confirmed that attitude (ATT) toward healthy food has a positive and significant effect on willingness to consume healthy food WCHBF (H6) (coefficient 0.169, *p* = 0.004). This suggests that a positive attitude is a significant predictor of consumption behavior. Likewise, perceived behavioral control (PBC) has a positive effect on willingness to consume healthy food (WCHBF) (H7): (coefficient 0.168, *p* = 0.005). This reflects that people who feel they have control over the consumption of these products are more likely to consume them. It has also been found that self-identity (SI) has a positive influence on the willingness to consume branded healthy food (WCHBF) (coefficient 0.116, *p* = 0.022). However, its impact is weaker compared to other variables. Moral norms (MN) have a positive and significant effect on the willingness to consume WCHBF branded healthy food (H9): (coefficient 0.162, *p* = 0.007), indicating that ethical values play a relevant role in this behavior.

Finally, it can be stated that MHLs are a key variable influencing all mediating variables, and directly WCHBF, underlining its importance in the model. ATT and PBC significantly impact WCHBF, reinforcing their relevance in healthy consumption decision-making. Although SI and MN have lower impacts, they also contribute to predicting behavior. This highlights the influence of ethical and identity factors on branded healthy food consumption. This comprehensive analysis confirms the validity of the proposed model in explaining the willingness to consume healthy branded food.

Table 7 shows how the variables personal identity (SI), perceived behavioral control (PBC), moral norms (MN), and attitude (ATT) act as mediators between healthy lifestyle motivators (MHLs) and a willingness to consume healthy brands of food (WCHBF). Personal identity (SI) significantly mediates the relationship between MHLs and WCHBF (O = 0.070, *p* = 0.024). This implies that the perception of healthy food as part of personal identity strengthens the relationship between these motivators and consumption. Likewise, PBC significantly mediates the relationship between MHLs and WCHBF (O = 0.101, *p* = 0.006). This suggests that consumers’ perceived control over their consumption decisions is essential in how healthy lifestyle motivators influence their behavior. MNs also act as mediators (O = 0.101; *p* = 0.008), indicating that ethical values and the perception of what is “right” reinforce the link between MHLs and WCHBF. ATT has the most considerable mediating effect among all variables (O = 117; *p* = 0.005). This highlights the importance of a positive attitude toward healthy food and how healthy lifestyle motivators lead to consumption. Finally, it can be stated that all mediating variables have a significant impact on the relationship between MHLs and WCHBF, although ATT stands out as having the strongest effect. The mediating variables reflect different dimensions of consumer behavior:

ATT: Evaluative perspective (emotional and cognitive).

PBC: Perception of personal capability.

MN: Ethical and social factors.

SI: Relationship to personal identity.

This analysis underscores that the model is multifaceted and that strategies to promote healthy consumption must address attitudinal aspects and perceptions of control, identity, and ethical norms.

## 5. Discussion

This study provides significant evidence regarding the influence of healthy lifestyle motivators (MHLs) on the willingness to consume healthy food brands (WCHBF) in the context of emerging markets, particularly in Peru. This study also analyzes this relationship through an integrative model that encompasses various psychological factors, such as attitude toward healthy food brands (ATT), perceived behavioral control (PBC), self-identity (SI), and moral norms (MN).

Initial results indicate that MHLs significantly influence WCHBF (β = 0.135, *p* = 0.004). This finding aligns with Chilón-Troncos et al.’s [23] finding that health perceptions and beliefs influence healthy food choices. However, the present study deepens this understanding by explicitly analyzing the role of healthy brand products in an emerging market context and a population predominantly with higher education. The favorable relationship between these motivators and consumer predisposition toward healthy products indicates a more intricate dynamic than those reported in previous studies, where individuals with more pronounced healthy lifestyle motivations exhibit a greater tendency toward health-focused brand products. This finding also extends the results of prior research, such as those by Qi and Ploeger [17] and Schubert [18], who determined that healthy lifestyle orientations affect the product choices consumers perceive as beneficial to health without considering specifically positioned brands. Nevertheless, the present study demonstrates that this relationship also manifests in a brand that consumers consider healthy. Additionally, the study by Nolan et al. [24] offers supplementary context for this study’s findings, identifying that healthy lifestyle considerations promote the choice of products that align with these values, promoting more nutritious dietary options.

The intensity of the relationship between MHLs and WCHBF constructs identified in this study can be understood through various sample-specific contextual factors. The demographic composition, predominantly young (91.97% aged 18–30 years) and university-educated (89.23% at the undergraduate level), provides information about the functioning of health motivators in this population segment where health care and healthy consumption awareness derived from education are likely factors that influenced the result. That is, young university students, especially those in health disciplines (69.57% in Health Sciences), tend to possess greater nutritional knowledge compared to the general population. Additionally, this demographic group is at a decisive phase in forming life habits, making them potentially more receptive to health messages and more inclined to convert health motivations into effective consumption decisions.

Another significant aspect to consider about the findings is that they come from an emerging market context, where healthy food consumption patterns and motivational factors may differ from those observed in more developed markets analyzed in previous research [17,18,23]. Therefore, even in emerging markets, health consciousness is fundamental in selecting certain food products and the brands that manufacture or market them, especially among young consumers with a higher level of education.

The findings reveal that MHLs exert a significant influence on multiple psychological factors. Thus, the analysis shows a strong impact on ATT (β = 0.692, *p* < 0.001), the most pronounced effect among all relationships examined. This finding significantly extends the results obtained by Salleh and Noor [9], who found a more modest relationship between health motivators and attitudes toward functional food. The difference in effect magnitude could be attributed to the present study examining brands positioned explicitly as healthy, while previous research analyzed functional products in general. This study also contributes to and extends the findings of Tudoran et al. [10] and Requero et al. [11], by demonstrating that health motivators influence attitude formation and have a powerful effect when dealing with brands with an established healthy identity.

Regarding PBC, the results show a significant effect (β = 0.599, *p* < 0.001) that remains important, although lower than the effect on ATT. This finding contrasts with the results of Prats-Arimon et al. [31] and Melnyk et al. [32], who found more moderate effects of health motivators on behavioral control. The difference could be explained by the focus on healthy brands, which could be perceived as facilitators of control over eating behavior. The results also extend the findings of Banerjee and Ho [33], by demonstrating that health motivators relate to behavioral control and contribute significantly to its development.

Regarding the influence of MHLs on SI, this study reveals a significant effect (β = 0.601, *p* < 0.001) that considerably contributes to previous findings. While Quaye et al. [39] and Yun and Silk [40] identified a general relationship between health motivators and self-identity, the present study demonstrates that this relationship is particularly notable in the context of healthy brands. The result also deepens the understanding proposed by Kirk and Tinning [41], suggesting that health motivators not only influence identity formation but can be a determining factor in how consumers integrate healthy brands into their self-concept. The results also reveal a significant influence of MHLs on MN (β = 0.623, *p* < 0.001), a finding that substantially expands the existing literature. While Honkanen et al. [46] found that health motivators can be internalized as moral values, the present study demonstrates that this internalization is particularly strong in the context of healthy brands. Furthermore, the results align with the findings of Arvola et al. [49], revealing that health motivators not only influence the formation of moral norms but have an effect comparable in magnitude to their influence on other psychological factors.

The intensity of these relationships reveals an interesting result: MHLs appear to have a more pronounced effect on evaluative aspects (attitude) than on control and normative aspects. This differentiation in effect magnitude provides new perspectives on how health motivators operate through different psychological mechanisms, thus contributing to a more nuanced understanding of their role in healthy food consumer behavior.

Another group of results analyzed the relationships between psychological factors and WCHBF, revealing significant influences among the constructs. For instance, ATT shows a positive and significant effect on WCHBF (β = 0.169, *p* = 0.004), supporting previous findings in various food market contexts. This finding aligns with the conclusions of Küster-Boluda and Vidal-Capilla [3], who found a similar relationship in the context of functional food. However, the effect magnitude in the present study is more moderate, revealing that, in the context of healthy brands, ATT operates alongside other equally important factors that influence WCHBF. This observation aligns with findings from various studies [51,52,53,54], although it deepens the understanding of how ATT explicitly influences the context of brands perceived to be healthy. Similarly, PBC significantly influences WCHBF (β = 0.168, *p* = 0.005), a finding that extends the existing literature. While previous studies [59,60,61,62] have identified a general relationship between perceived control and healthy food consumption. The present study reveals that this relationship holds explicitly for healthy brands. The effect magnitude is comparable to that of ATT, indicating that both factors have similar importance in determining consumption willingness. This finding is particularly relevant as it contrasts with previous studies where perceived behavioral control showed effects of different magnitudes.

Regarding SI, the results indicate a significant, albeit more modest, influence on WCHBF (β = 0.116, *p* = 0.022). This finding extends the results of Cavallo et al. [1] and Ateş [65], who found links between health-conscious identity and healthy food purchasing. However, the lower effect magnitude reported in the present study indicates that SI might play a more subtle role in the context of healthy brands than other psychological factors. This difference could be attributed to this study focusing on a specific brand rather than general categories of nutritious food. Additionally, MN significantly influences WCHBF (β = 0.162, *p* = 0.007), with a magnitude comparable to ATT and PBC. This result is fascinating as it augments the literature regarding previous research [71,72,73] that has identified the importance of ethical values in healthy food consumption decisions. The similarity in effect magnitude with other psychological factors suggests that moral considerations are as important as the more traditionally studied factors in consumer behavior.

Collectively, the similarity in effect magnitudes of ATT, PBC, and MN (β ≈ 0.16–0.17), along with the more moderate effect of SI (β = 0.116), reveals that the willingness to consume healthy food brands results from a balanced interaction among different psychological factors rather than the dominance of a particular factor. This finding contributes significantly to understanding how consumers make decisions about healthy food brands, suggesting that marketing strategies and public health interventions could be more effective if they simultaneously address multiple psychological factors.

Regarding mediation effects, the results reveal that psychological factors act as significant mediators between MHLs and WCHBF. Specifically, ATT demonstrates the strongest mediating effect (β = 0.117, *p* = 0.005), which extends the findings of previous studies that have examined attitude primarily as a direct predictor of behavior. This result evidences that health motivators operate partly through forming positive attitudes toward healthy food, a mechanism not explicitly identified in previous research [51,52,53,54]. Similarly, PBC and MN exhibit mediating effects of similar magnitude (β = 0.101, *p* = 0.006 and β = 0.101, *p* = 0.008, respectively). This parity in mediation effects is particularly interesting, as it reveals that both control aspects and moral values are equally crucial in translating health motivations into a willingness to consume. This finding extends our previous understanding [59,60,61,62,71,72,73] by demonstrating that these factors directly influence behavior and serve as mechanisms through which health motivators exert influence. Similarly, SI presents the most modest yet significant mediating effect (β = 0.070, *p* = 0.024). This result differs from previous studies [1,65] that have emphasized the direct role of identity in consumer behavior. The lower magnitude of the mediating effect may indicate that, although SI is a valid mechanism, it might be less crucial than other factors in translating health motivations into a willingness to consume healthy food brands. The general pattern of mediation effects reveals a hierarchy of psychological factors. ATT is the most potent mediator, followed by PBC and MN in equal measure and SI. This hierarchical structure provides new perspectives on how health motivators translate into a willingness to consume through different psychological pathways.

### Implications

This study underscores the pivotal role of psychological motivators—such as self-identity, moral norms, and perceived behavioral control—in influencing healthy food consumption. Marketers can leverage these insights to craft campaigns that resonate emotionally with consumers by emphasizing these motivators. Specifically, aligning brand identity with personal values and promoting the ethical and identity-affirming aspects of healthy eating can drive brand loyalty and improve purchasing behavior.

For policymakers, the findings highlight the importance of integrating moral and ethical considerations into public health initiatives. Campaigns should frame healthy eating as a socially responsible choice that contributes to sustainability and communal well-being. Programs to improve access to and the affordability of healthy food should also enhance perceived behavioral control, empowering consumers to make healthier dietary choices.

This study also offers actionable insights for educators and institutions involved in nutritional awareness. By emphasizing the alignment of healthy eating with personal and social values, these initiatives can foster a deeper and more sustained impact on dietary behavior.

From a research perspective, this study contributes to understanding consumer psychology in the context of health-oriented markets in developing economies. It allows further exploration into cross-cultural applications of these findings and longitudinal studies to assess behavioral changes over time.

## 6. Conclusions

This study aimed to analyze the impact of motivators toward a healthy lifestyle (MHLs) on a Peruvian brand’s willingness to consume healthy food (WCHBF). For this purpose, attitude toward healthy eating (ATT), perceived behavioral control (PBC), self-identity (SI), and moral norms (MN) were considered as study variables.

According to the results and hypothesis tests, MHLs significantly influence WCHBF. This indicates that MHLs play a determining role in orienting people toward adopting a healthy diet.

In the present investigation, the effects of ATT, PBC, SI, and MN variables on WCHBF were highly significant. These findings indicate that these variables play a crucial role in the perception of consumers’ importance and personal value of healthy food, influencing their consumption decisions. This phenomenon can be interpreted as an inclination toward choosing healthy products, which in turn facilitates the identification of brands that meet the needs of this consumer segment.

The results indicated that these variables function as mediators between MHLs and WCHBF. This suggests that the connection between these motivators and consumption is strengthened, as consumers recognize that purchasing decisions are crucial in influencing MHLs’ behavior. This indicates that ethical values and perceptions of what is “right” consolidate the link between MHLs and WCHBF, underscoring the relevance of a favorable attitude toward healthy food and how healthy lifestyle motivators encourage consumption. This analysis emphasizes that the model is multifaceted and that strategies to promote healthy consumption should consider attitudinal aspects and perceptions of control, identity, and ethical norms.

In marketing campaigns, companies offering healthy products should mention MHLs so that consumers can identify their characteristics. This will allow consumers to recognize the brand, which will generate trust, reliability, and consistency in the quality of the healthy products offered by the company. Consumers who place their trust in a corporate brand perceived as a provider of health products and feel satisfied with it will maintain their preference for that brand.

### Limitations and Future Research

While this study provides valuable insights into the impact of motivators as determining factors on the intention to purchase healthy food in a developing nation, it is not without limitations. Firstly, the cross-sectional design of the research constrains the ability to establish definitive causal relationships among the variables under examination. Although the correlations between healthy lifestyle motivators, attitudes, perceived behavioral control, self-identity, moral norms, and a willingness to consume healthy food provide a helpful overview, they do not facilitate conclusions about causality or the evolving nature of these relationships over time.

Secondly, this study’s focus was limited to Lima, potentially overlooking the attitudes and behaviors of consumers in other regions of the country or other countries and continents. Regional cultural, economic, and social disparities can significantly influence perceptions of motivators and, consequently, the intention to purchase healthy products. This limitation is particularly salient considering the demographic profile of the participants, the majority of whom are under 30 and predominantly university students or graduates, a demographic likely to recognize the importance of healthy eating for overall well-being.

Thirdly, the sample was obtained through non-probabilistic convenience sampling, which may introduce selection biases and restrict the generalizability of the findings. Despite the survey reaching a broad audience via social media, the participants may not adequately represent the broader population of healthy food consumers in Peru.

For subsequent research endeavors, we suggest several areas of exploration. Firstly, replicating this study across various social contexts and geographical regions would enhance the representativeness and generalizability of the findings. Comparative analyses across different countries could uncover significant variations in consumer perceptions and behaviors, offering a more nuanced understanding of the healthy food market beyond the confines of Peru. Expanding research efforts to include South America and other regions with analogous characteristics, such as parts of Asia and Africa, could be beneficial.

Additionally, longitudinal studies are recommended to evaluate how the relationships among motivators, attitudes, moral norms, perceived behavioral control, self-identity, and purchase intention evolve. Such an approach would deepen our understanding of these relationships’ causal dynamics and stability concerning healthy food consumption. Complementing this quantitative framework with qualitative studies could provide richer insights into individual behaviors, contextual factors, and their implications for purchasing intentions.

Furthermore, future investigations might explore additional factors influencing purchase intentions for healthy food, building upon Janz and Becker and the basic elements of the Health Belief Model [88]. Other moderating variables, such as health awareness and nutritional literacy, could also be included in the model. Incorporating a broader set of variables would allow us to understand better the determinants of willingness to consume healthy food brands.

## Figures and Tables

**Figure 1 foods-14-00125-f001:**
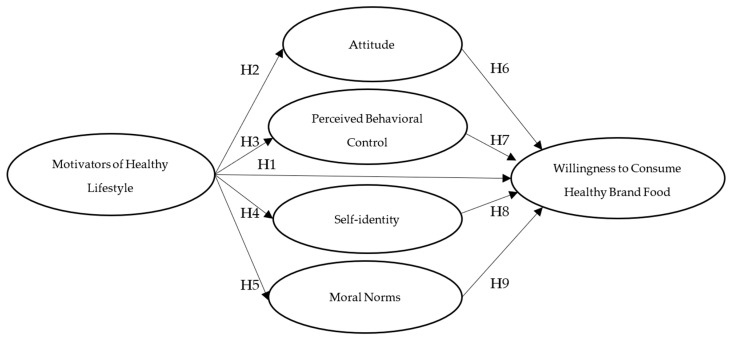
Conceptual model.

**Figure 2 foods-14-00125-f002:**
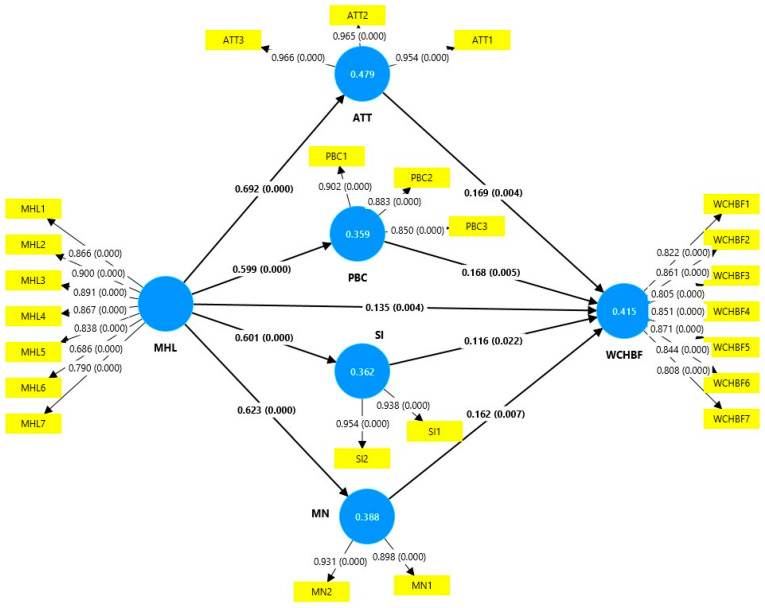
Structural model. HLB = healthy lifestyle, ATT = attitude, PBC = perceived behavioral control, SI = self-identity, MN = moral norms, and WCHBF = willingness to consume healthy food.

**Table 1 foods-14-00125-t001:** Sociodemographic data.

Sex	*n*	%
Man	205	35.04%
Women	380	64.96%
Total	585	100.00%
Age	*n*	%
Young (18 to 30 years old)	538	91.97%
Adult (31 to 60 years old)	47	8.03%
Total	585	100.00%
Area of Study	*n*	%
Health Sciences	407	69.57%
Business Studies	133	22.74%
Other	45	7.69%
Total	585	100.00%
Marital status	*n*	%
Single	544	92.99%
Married	35	5.98%
Other	6	1.03%
Total	585	100.00%
Academic level	*n*	%
Undergraduate	522	89.23%
Postgraduate	38	6.50%
School	18	3.08%
Technical	7	1.20%
Total	585	100.00%
Religion	*n*	%
Adventist	475	81.20%
Evangelical	14	2.39%
Catholic	90	15.38%
Agnostic	6	1.03%
Total	585	100.00%

**Table 2 foods-14-00125-t002:** Reliability and validity analysis of the measurement model.

Variable	Code	Loading	Cronbach’s Alpha	Composite Reliability (rho_a)	Average Variance Extracted (AVE)
Attitude	ATT1	0.954	0.959	0.959	0.925
	ATT2	0.965			
	ATT3	0.966			
Healthy Lifestyle Motivators	MHL1	0.866	0.927	0.934	0.700
	MHL2	0.900			
	MHL3	0.891			
	MHL4	0.867			
	MHL5	0.838			
	MHL6	0.686			
	MHL7	0.790			
Moral Norms	MN1	0.898	0.806	0.825	0.837
	MN2	0.931			
Perceived Behavioral Control	PBC1	0.902	0.852	0.859	0.772
	PBC2	0.883			
	PBC3	0.850			
Self-identity	SI1	0.938	0.883	0.896	0.895
	Sl2	0.954			
Willingness to Consume Healthy Brand Food	WCHBF1	0.822	0.929	0.933	0.702
	WCHBF2	0.861			
	WCHBF3	0.805			
	WCHBF4	0.851			
	WCHBF5	0.871			
	WCHBF6	0.844			
	WCHBF7	0.808			

**Table 3 foods-14-00125-t003:** Fornell–Larcker criterion.

	ATT	MHL	MN	PBC	SI	WCHBF
Attitude (ATT)	0.962					
Healthy Lifestyle Motivators (MHLs)	0.692	0.837				
Moral Norms (MN)	0.713	0.623	0.915			
Perceived Behavioral Control (PBC)	0.688	0.599	0.729	0.879		
Self-identity (SI)	0.620	0.601	0.695	0.675	0.946	
Willingness to Consume Healthy Brand Food (WCHBF)	0.566	0.524	0.570	0.562	0.528	0.838

**Table 4 foods-14-00125-t004:** Cross loadings.

ITEMS	ATT	MHL	MN	PBC	SI	WCHBF
ATT1	0.954	0.670	0.694	0.670	0.607	0.538
ATT2	0.965	0.662	0.684	0.656	0.586	0.546
ATT3	0.966	0.665	0.679	0.659	0.596	0.549
MHL1	0.595	0.866	0.552	0.534	0.520	0.427
MHL2	0.670	0.900	0.575	0.561	0.536	0.477
MHL3	0.636	0.891	0.557	0.521	0.480	0.468
MHL4	0.631	0.867	0.554	0.551	0.537	0.460
MHL5	0.571	0.838	0.490	0.478	0.478	0.438
MHL6	0.372	0.686	0.417	0.375	0.481	0.359
MHL7	0.534	0.790	0.486	0.464	0.496	0.431
MN1	0.545	0.487	0.898	0.613	0.591	0.501
MN2	0.743	0.641	0.931	0.714	0.674	0.540
PBC1	0.709	0.586	0.700	0.902	0.607	0.533
PBC2	0.606	0.503	0.632	0.883	0.571	0.456
PBC3	0.485	0.482	0.583	0.850	0.599	0.488
SI1	0.553	0.532	0.627	0.626	0.938	0.455
Sl2	0.616	0.601	0.684	0.649	0.954	0.540
WCHBF1	0.573	0.494	0.559	0.535	0.469	0.822
WCHBF2	0.473	0.450	0.493	0.509	0.430	0.861
WCHBF3	0.405	0.390	0.404	0.441	0.415	0.805
WCHBF4	0.529	0.491	0.497	0.491	0.476	0.851
WCHBF5	0.471	0.437	0.472	0.469	0.425	0.871
WCHBF6	0.426	0.421	0.469	0.432	0.484	0.844
WCHBF7	0.409	0.364	0.424	0.395	0.387	0.808

**Table 5 foods-14-00125-t005:** Heterotrait–monotrait ratio (HTMT)—Matrix.

	ATT	MHL	MN	PBC	SI	WCHBF
Attitude (ATT)						
Healthy Lifestyle Motivators (MHLs)	0.728					
Moral Norms (MN)	0.800	0.711				
Perceived Behavioral Control (PBC)	0.755	0.668	0.871			
Self-identity (SI)	0.671	0.665	0.817	0.776		
Willingness to Consume Healthy Brand Food (WCHBF)	0.593	0.559	0.652	0.625	0.578	

**Table 6 foods-14-00125-t006:** Hypothesis testing.

H	Hypothesis	Original Sample (O)	Sample Mean (M)	*p* Values	Decision
H1	MHL → WCHBF	0.135	0.135	0.004	Accepted
H2	MHL → ATT	0.692	0.693	0.000	Accepted
H3	MHL → PBC	0.599	0.600	0.000	Accepted
H4	MHL → SI	0.601	0.602	0.000	Accepted
H5	MHL → MN	0.623	0.624	0.000	Accepted
H6	ATT → WCHBF	0.169	0.169	0.004	Accepted
H7	PBC → WCHBF	0.168	0.170	0.005	Accepted
H8	SI → WCHBF	0.116	0.117	0.022	Accepted
H9	MN → WCHBF	0.162	0.161	0.007	Accepted

**Table 7 foods-14-00125-t007:** Mediation effects testing.

	Original Sample (O)	Sample Mean (M)	Standard Deviation (STDEV)	T Statistics (|O/STDEV|)	*p* Values
MHL → SI → WCHBF	0.070	0.070	0.031	2.252	0.024
MHL → PBC → WCHBF	0.101	0.102	0.037	2.728	0.006
MHL → MN → WCHBF	0.101	0.100	0.038	2.650	0.008
MHL → ATT → WCHBF	0.117	0.117	0.041	2.839	0.005

## Data Availability

The original contributions presented in the study are included in the article, further inquiries can be directed to the corresponding author.

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
