# Peer review of "Healthy Lifestyle Motivators of Willingness to Consume Healthy Food Brands: An Integrative Model"

_foods, 2025, doi:10.3390/foods14010125_

Round 1

Reviewer 1 Report

Comments and Suggestions for Authors

Report on the manuscript foods-3299249 entitled: Healthy Lifestyle Motivators as Determining Factors in Healthy Food Consumption: A Study on Self-Identity, Attitudes, and Moral Norms.

-          The population described in Table 1 cannot be considered pondered regarding sex/gender, marital status, age, religion, academic level, etc.

How were such differences addressed regarding the statistical analysis?

It has been described in the Conclusions that “…attitude, behavior, self-identity, and morality are key factors that generate the importance and personal value of each of the consumers…”. Nevertheless, such deductions cannot be considered since the respective populations lack uniformity.

In fact, the basis of the manuscript should be reviewed.

Considering Table 1, the statement of “healthy lifestyle motivators (HLM) on willingness to consume healthy foods (WCHBF)…” would not be appropriate due to the unfitted populations.

-          L. 310-315. It is odd to consider R2 values of 0.10 as acceptable and from 0.25 as moderate and > 0.50 as explanatory.

Further explanation and description must be provided.

Actually, R2 values lower than 0.65 are widely considered as useless to the industry.

-          Figures 1 and 2 (and corresponding Tables): Naming and abbreviations should be double-checked. It seems to be a mix of PBC and PCB, NM and MN, MHL meaning?

-          Results section (L. 317-345) should be improved.

Further in-depth descriptions of the results should be considered.

-          Discussion section (L. 356-421).

Lacking adequate comparisons and critical analysis.

Author Response

Dear Reviewer,

We extend our sincere gratitude for your insightful comments, which have been invaluable in enhancing the quality of our manuscript. Your thoughtful feedback has contributed significantly to refining our work, and we have made concerted efforts to address each of your suggestions.

We are optimistic that this revised version of the paper now meets the anticipated standards for publication in this esteemed journal. Below is a comprehensive list of responses addressing your comments and suggestions.

Thank you once again for your time and expertise.

Best regards,

Reviewer 2 Report

Comments and Suggestions for Authors

The title is interesting

The abstract is well designed and appropriate

The research problem and motive are well addressed

Please add at the end of the introduction section the importance of Peru as a context of your study

You need to strengthen the theoretical background of your study.

Literature review and hypotheses justification are adequately presented

The sample size is adequate

You need to address the whole population as well

The employed data analysis technique (PLS-SEM) is appropriate

The convergent validity results are satisfactory

The GOF for PLS model is adequate

Please report and elaborate on the cross loadings and HTMT results for the scale discriminant validity

The PLS-Sem inner model results are good

Please strengthen the discussion and implications sections.

Author Response

(The authors gave the same response as above.)

Reviewer 3 Report

Comments and Suggestions for Authors

It appears that G*Power does not currently support SEM calculations, so I am unsure how the authors obtained the minimum required sample size using this software, as mentioned in line 266.

Table 1 could be formatted as a single column instead of two, which would make it clearer.

Since all the independent variables come from the same scale, this implies that most of the variables in the model have already been measured in prior studies, with the authors seemingly only changing the dependent variable. This approach may limit the novelty of the study. It could be beneficial to further explore mediation effects, moderation effects, or employ fsQCA to analyze the necessity and configuration of conditions, which would provide greater depth and completeness to the research.

Please provide the full questionnaire and raw data. The AVE values in this study are exceptionally high, which is surprising; I am curious about the types of questions that could lead to such data.

Some of the references are outdated; please cite more recent literature from the past five years.

Author Response

(The authors gave the same response as above.)

Round 2

Reviewer 1 Report

Comments and Suggestions for Authors

Report on the manuscript foods-3299249 entitled: Healthy Lifestyle Motivators of Willingness to Consume Healthy Brand Food: An Integrative Model.

The manuscript has been improved.

Some issues before being considered for publication:

-        -  Please, use dot (“.”) instead of comma (“,”) to indicate decimal point. Review Tables 4, 5, 7.

-       -   I cannot understand the need of Table 4 when the same values are shown in Figure 2.

Author Response

Dear Reviewer 1,

Thank you for your valuable feedback on our manuscript "Healthy Lifestyle Motivators of Willingness to Consume Healthy Brand Food: An Integrative Model." We have carefully addressed your comments as follows:

  1. Decimal Points: We have revised Tables 4, 5, and 7 to use dots (.) instead of commas (,) as decimal separators to maintain consistency with international standards.
  2. Regarding Table 4 and Figure 2: While we acknowledge the overlapping information between Table 4 and Figure 2, we have maintained Table 4 in response to Reviewer 2's specific request to explicitly include Tables 4 and 5 in the results section, even though Figure 2 visualizes these data points. This decision accommodates the reviewer's feedback while ensuring a comprehensive presentation of our results. We hope these revisions meet your requirements. Please let us know if any additional modifications are needed.

Best regard,

Authors

Reviewer 2 Report

Comments and Suggestions for Authors

can accept in present form 

Author Response

Dear Reviewer
Thank you very much for your comments which helped to improve the quality of our work.

Best regards

Reviewer 3 Report

Comments and Suggestions for Authors

I have no further suggestions. Congratulations!

Author Response

(The authors gave the same response as above.)
